# Wind Direction Estimation Using SCADA Data with Consensus-Based Optimization

Jennifer Annoni[1], Christopher Bay[1,2,3], Kathryn Johnson[1,2], Emiliano Dall'Anese[3], Eliot Quon[1], Travis Kemper[1], and Paul Fleming[1]

[1]National Renewable Energy Laboratory, Golden, CO, 80401, USA
[2]Colorado School of Mines, Golden CO, 80401, USA
[3]University of Colorado - Boulder, Boulder, CO, 80309, USA

*Correspondence to:* Jennifer Annoni (jennifer.annoni@nrel.gov)

**Abstract.**

Wind turbines in a wind farm typically operate individually to maximize their own performance and do not take into account information from nearby turbines. To enable cooperation to achieve farm-level objectives, turbines will need to use information from nearby turbines to optimize performance, ensure resiliency when other sensors fail, and adapt to changing local conditions.

A key element of achieving a more efficient wind farm is to develop algorithms that ensure reliable, robust, real-time, and efficient operation of wind turbines in a wind farm using local sensor information that is already being collected, such as supervisory control and data acquisition (SCADA) data, local meteorological stations, and nearby radars/sodars/lidars. This article presents a framework for developing a cooperative wind farm that incorporates information from nearby turbines in real-time to better align turbines in a wind farm. SCADA data from multiple turbines can be used to make better estimates

of the local inflow conditions at each individual turbines. By incorporating measurements from multiple nearby turbines, a more reliable estimate of the wind direction can be obtained at an individual turbine. The consensus-based approach presented in this paper uses information from nearby turbines to estimate wind direction in an iterative way rather than aggregating all the data in a wind farm at once. Results indicate that this estimate of the wind direction can be used to improve the turbine's knowledge of the wind direction. This estimated wind direction signal has implications for potentially decreasing dynamic

yaw misalignment, decreasing the amount of time a turbine spends yawing due to a more reliable input to the yaw controller, increasing resiliency to faulty wind-vane measurements, and increasing the potential for wind farm control strategies such as wake steering.

## 1   Introduction

The wind industry continues to seek methods to decrease the levelized cost of energy (LCOE) by using advances in science,

engineering, and computation (Lindenberg (2009)). Control systems have the potential to contribute to this LCOE reduction by incorporating local measurements and optimizing large wind farms in real time. Specifically, control systems can be used to achieve a cooperative wind farm, which includes turbines that self-organize into groups, monitor their status and the status of other turbines, and control their performance to maximize the economic and reliable performance of a large-scale wind farm. A

framework for cooperative wind farms can be achieved by representing a wind farm as a network of wind turbines. The network topology can be used to advance the state of the art in wind farm controls in topics ranging from distributed optimization and control to fault detection and short-term forecasting. In this sense, wind farms can take advantage of the network topology to implement scalable, reconfigurable, and resilient control strategies in real time. The approach presented in this article is an iterative algorithm that takes advantage of the topology of a wind farm and incorporates local measurements from nearby turbines to determine the wind direction at an individual turbine. Improving the wind direction measurement at the turbine can reduce unnecessary yaw movements and minimize dynamic yaw misalignments.

Currently, turbines typically rely on wind vanes and anemometers mounted on the back of the nacelle to provide measurements to their controllers. Some turbine manufacturers have wind speed and wind direction estimators to correct for these errors based on individual turbine measurements. Individual measurements, on their own, can be unreliable due to the complex flow created as the wind passes through the rotor, preventing accurate inputs into the individual turbine yaw controller. One way to address noisy wind direction information is to mount a forward-facing lidar on the nacelle to detect the wind in front of the turbine (Fleming et al. (2014b); Schlipf et al. (2013); Scholbrock et al. (2013); Simley et al. (2014)). In addition, meteorological (met) towers can be used to characterize the inflow; however, most turbines do not have dedicated met towers and the wind direction can vary across a wind farm due to variable meteorological conditions and topography. Other remote sensing techniques have been proposed as well including radar, lidar, sodar, etc. (Peña et al. (2015); Barthelmie et al. (2016)). However, they all require additional sensing equipment and integration into turbine controllers.

This paper describes a distributed optimization-based method to reliably estimate the wind direction across a wind farm even when faults and/or biases are present. Distributed optimization and control theory provide a framework for efficient computation of large systems, especially systems with network topologies. These types of optimization strategies have been used for multiagent systems, such as unmanned aerial vehicles and robots, and can be used to coordinate subsystems to interact with their larger environment (Zhu and Martínez (2015); Ferrari et al. (2016); Movric and Lewis (2014); Shamma (2008)). Distributed optimization has also been considered in the wind farm controls literature (Marden et al. (2012); Spudić et al. (2015)). However, complex aerodynamic interactions and large timescales make this a challenging problem. For example, a distributed optimization framework for wind farm controls has been presented in (Soleimanzadeh et al. (2013)). However, solving this problem becomes computationally complex as the system grows because of the number of turbines and larger flow domains. A limited-communication distributed model predictive controller designed to track a power reference signal is described in Bay et al. (2018); this algorithm uses a simplified linearized wake model to describe turbine interactions, allowing for scalability. Because this method requires a linear model, this method is difficult to extend to power maximization or load minimizations where the objective functions are highly nonlinear.

Consensus-based algorithms are a specific class of optimization algorithms that have the potential to accommodate sensor errors caused by failure, miscalibration, and noise by assuming that turbines experience wind inflow directions that share similar characteristics with that of their neighbors. Although consensus optimization is an active area of research for many applications, especially in multiagent systems, its use in wind farm applications is relatively new. A few studies have used consensus-based approaches for various problems in wind energy. In one such study, a dynamic average consensus estimator is

used in Ebegbulem and Guay (2017) to estimate an overall cost function for turbines communicating via an undirected network where the goal is to maximize total wind farm power production. Similar power maximization approaches using consensus-based approaches for an undirected graph can be found in Wang et al. (2017); Gionfra et al. (2017). Finally, the research of Baros and Ilic (2017) allows turbines to self-organize using torque control and storage to regulate total wind farm power output.

5       This article presents a consensus-based distributed optimization algorithm for reliably calculating wind direction at a wind turbine using only supervisory control and data acquisition (SCADA) data from the turbines in the wind farm. This wind direction estimate can be used as an input to a turbine yaw controller, facilitate wake steering wind farm control (Fleming et al. (2014a)) and other forms of wind farm control, inform operations management, and provide condition monitoring. It is important to note that this approach requires no additional sensing information. This algorithm is based on the work presented

in Hallac et al. (2015) and is solved using alternating direction method of multipliers (Boyd et al. (2011)). Details of this algorithm are found in Section 3.1. This method is also compared to alternative approaches to estimating wind direction such as averaging. The results are discussed in Section 5. A key contribution of this article is demonstrating this algorithm on wind farm SCADA data where the wind direction varies across the wind farm. For proprietary reasons, all the data has been normalized and only a subsection of the wind farm is shown (see Section 5.2). Results of this approach are compared with a

sodar on-site and are shown in Section 6. The results indicate that this approach provides a robust measurement of the wind direction at each turbine. Finally, Section 7 provides some conclusions and suggestions for future work.

## 2   Wind Farm as a Network

In this framework, turbines can take advantage of data from nearby turbines to make more informed decisions about individual or farm-level operation. This framework is scalable, reconfigurable, and can be extended to include additional sensors incorpo-

rating nearby, relevant measurements from other turbines, meteorological instruments, and mobile sensors. Identifying a graph or network topology is important for incorporating local information and taking advantage of the structure of the wind farm to perform real-time optimization. The network connections can be based on proximity or based on physical interactions, i.e. aerodynamic interactions (wakes). This study uses proximity to define the connections between turbines. Finally, this approach can solve local optimization problems and allows for local variations experienced in a wind farm. The wind farm can be mod-

eled as an undirected or a directed network where turbines communicate with nearby turbines. Turbines in a wind farm can be considered the nodes and the edges are established communication between nearby turbines. Information is communicated across these edges to determine local atmospheric conditions—such as wind direction or wind speed—at a particular turbine.

    It is important to note that although turbines typically communicate with a central computer to record SCADA data, an undirected network is used to determine which turbines to include while computing a local optimization at a particular turbine.

This topology is designed to take advantage of temporal and spatial structures in a wind farm. For example, a turbine on the western most edge of the wind farm can be experiencing a different wind speed/direction than a turbine on the eastern edge of the wind farm that is several kilometers away. This study uses a nearest neighbor approach to define the network topology. Additional ways to characterize the network topologies will be explored in future work.

There are a few important things to note when determining the network topology in a wind farm. First, the local conditions can vary across a wind farm. The number of connections between each turbine can determine the variability obtained across the wind farm. For example, if every turbine is connected to every other turbine, the variability across the wind farm will be small and the turbines will all agree on one set of atmospheric conditions based on a consensus approach. In addition, the computation time will be high due to the number of communication exchanges per iteration of the consensus algorithm. However, if the turbine has no connections, the variation of the output of the consensus algorithm across the wind farm will be high and possibly unreliable due to sensor noise and miscalibration. Smaller groups of turbines can agree on local conditions and provide a reliable measurement that more accurately captures the variations across the wind farm.

## 3 Distributed Optimization for Real-Time Operation

A distributed approach can be used to solve an optimization problem that takes advantage of a system's network topology. In this particular case, a distributed optimization problem will be used to agree on wind direction across turbines. The problem can be decomposed such that each turbine can solve its own optimization problem, which incorporates information from connected turbines in the network topology. In other words, a few measurements from nearby turbines are used to solve the optimization problem rather than solving a centralized optimization problem that includes all measurements from all turbines. Trying to incorporate all measurements from all turbines can be problematic in terms of communication limits and computational complexity, potentially taking hours to solve. However, grouping the turbines in a wind farm based on distance provides a computationally efficient algorithm for optimizing a particular objective function. The approach used in this article is based on Hallac et al. (2015) where the objective function can be written as:

$$\underset{x_i}{\text{minimize}} \quad \underbrace{\sum_{i \in \mathcal{V}} f_i(x_i)}_{\text{node objective}} + \underbrace{\sum_{(j,k) \in \mathcal{E}} g_{jk}(x_j, x_k)}_{\text{edge objective}} \quad i = 1, ..., N_{turbs} \quad j \in \mathcal{N}(i) \tag{1}$$

where $f_i(x_i)$ is the objective function at turbine $i$, i.e., the node objective; $\mathcal{N}_i$, indicates the turbines connected to turbine $i$; $\mathcal{V}$ is a set of all nodes and $\mathcal{E}$ is a set of all edges in the graph; $x_i$ is the wind direction estimate at turbine $i$; and $g_{jk}(x_j, x_k)$ compares wind direction measurements between turbines $j$ and $k$ in $\mathcal{N}$ in the wind farm network, i.e., the edge objective. Each turbine is a node in $\mathcal{V}$ and the nearest turbines are connected by edges in $\mathcal{E}$. This is a generalized framework for estimating many different quantities throughout the wind farm such as power, wind speed, or wind direction. The next section describes a wind direction consensus example to illustrate the utility of this framework.

### 3.1 Wind Direction Consensus

This study uses a consensus-based approach to reliably determine the wind direction at every turbine considering both the individual turbine measurements and those of its nearest X neighbors. The SCADA data measurements recorded at each turbine are used to determine a reliable measurement of wind direction at every turbine. This approach allows the wind direction and

wind speed to vary across a wind farm. It is assumed that the wind directions recorded at the turbines are with reference to true north and that the wind direction varies smoothly across the wind farm.

For this problem, each turbine uses its own wind direction measurement, $\tilde{x}_{i,measure}$ as well as the wind direction measurement from the connected turbines to determine the local wind direction. First, the objective of the individual turbine $i$, i.e., the node objective, is to minimize the error between the wind direction measurement measured at turbine $i$ and the estimated wind direction, $x_i$, or:

$$f_i(x_i) = (\tilde{x}_{i,measure} - x_i - b_i)^2 + \alpha|b_i| \tag{2}$$

where $b_i$ is the measurement bias in the wind direction with respect to true north, and $\alpha$ enforces sparsity of these biases in the wind farm. A large $\alpha$ allows for very few biases to be identified among turbine measurements in the wind direction consensus algorithm. A small $\alpha$ allows for many biases to be identified. These biases can help identify faults in wind direction measurements. This approach allows the biases to be incorporated into the optimization, rather than be ignored, which can contribute to the overall wind direction consensus. In addition to the node objective, the edge objective incorporates information from nearby turbines to ensure a reliable measurement of the wind direction at an individual turbine. The edge objective can be written as:

$$g_{ij}(x_i, x_j) = w_{ij}|x_i - x_j| \tag{3}$$

where $w_{ij}$ is a weight placed on the connection between turbines $i$ and $j$, $x_i$ is the estimated wind direction at turbine $i$, and $x_j$ is the estimated wind direction at turbine $j$. The edge objective, $g_{ij}(x_i, x_j)$, minimizes the differences in estimated wind direction between neighboring turbines. In this paper, the weights, $w_{ij}$, in each cluster are weighted based on distance using a normal distribution, i.e. the information from turbines that are closer is weighted more than turbines that are farther away. Equations (2) and (3) are used by the optimization problem (1) for this wind direction consensus problem. Using clustering allows each subset optimization to be performed in parallel, further reducing computational time. An iterative approach is needed to solve this optimization problem and is detailed in the next section. This iterative approach provides a feedback mechanism that lends itself to additional benefits such as fault detection.

## 3.2 Alternating Direction Method of Multipliers

Alternating direction method of multipliers (ADMM) is a technique used to solve distributed optimization problems (Boyd et al. (2011)) such as (1). This algorithm is particularly useful in this case since each individual turbine solves its own optimization in parallel, communicates the solution to neighboring subsets, and iterates this process until the wind farm has converged and each node has reached a single value. In this study, each turbine determines its own local wind direction by only talking to its nearest neighbors, as indicated in Section 2. ADMM is used to solve a network optimization with connecting nodes to

determine a consensus between shared nodes such that:

$$\text{minimize} \quad \sum_{i}^{N_{turbs}} f_i(x_i, b_i) \quad + \quad \lambda \sum_{(j,k)\in\mathcal{E}} w_{jk} \|z_{jk} - z_{kj}\|_2 \tag{4}$$

$$\text{subject to} \quad x_i = z_{ij}, \quad j \in N(i) \tag{5}$$

where $z_{jk}$ are copies of $x$ at different nodes, i.e. $z_{j,k}$ is a copy of $x_j$ at turbine $k$ such that the wind farm reaches consensus of the wind direction across the wind farm, and $\lambda$ is a penalty term that enforces consensus. If $\lambda$ is very large, there will be total consensus. If $\lambda$ is very small, there will be no consensus among nodes. The operator $\|\cdot\|_2$ indicates the $L_2$ norm.

The distributed optimization problem is solved using ADMM by minimizing the augmented Lagrangian

$$\mathcal{L}_\rho(x, b, z, u) = \sum_{i\in\mathcal{V}} f_i(x_i, b) + \sum_{(j,k)\in\mathcal{E}} \lambda w_{jk} \|z_{jk} - z_{kj}\|_2 - (\rho/2)\left(\|u_{jk}\|_2^2 + \|u_{kj}\|_2^2\right) + (\rho/2)\left(\|x_j - z_{jk} + u_{jk}\|_2^2 + \|x_k - z_{kj} + u_{kj}\|_2^2\right) \tag{6}$$

where $u$ is the scaled dual variable and $\rho > 0$ is the penalty parameter that enforces the constraints on the problem, i.e. enforces $x_i = z_{i,j}$. The variables $x$, $b$, $z$, and $u$, are updated in serial. In this particular setup, the bias, $b_i$, does not have a $z$-update or $u$-update since the biases are only known to the individual turbines, i.e. they are not communicated to nearby turbines.

## 4  Alternative Methods for Estimating Wind Direction

In addition to a consensus-based approach, alternative methods can be used to determine an estimate of the wind direction given the measurements of nearby turbines in the wind farm that are less computationally expensive. This section presents three alternative methods that will be used for comparison to the consensus-based approach in the presence of a fault in one of the turbines. Results of the comparison will be given in Section 5.

### 4.1  Averaging

The first and simplest approach uses averaging across all of the turbines such that:

$$x_i = \frac{1}{N} \sum_{j}^{N} x_{j,measure} \tag{7}$$

where $x_i$ is the modified wind direction signal of turbine $i$, $N$ is the number of turbines in the wind farm, and $x_{j,measure}$ is the wind direction measured at turbine $j$. The main limitation to this approach is that it provides only one wind direction across the entire farm. It is common for the wind direction to vary across a large wind farm, so averaging is likely to give poor results for some turbines. Also, averaging is expected to perform worse in the presence of faults.

## 4.2 Weighted Averaging

The next approach that was examined was a weighted average of the wind directions at each turbine such that:

$$x_i = \sum_j^N w_{ij} x_{j,measure} \tag{8}$$

where $w_{ij}$ is a weight based on the distance between turbines $i$ and $j$. In this case, we used a normal distribution that weights turbines that are closer to turbine $i$ with a higher weight and turbines that are farther away from turbine $i$ with a lower weight. This method allows for changes in wind direction across the wind farm, which is likely to be an improvement over simple farm-wide averaging. However, faulty sensor cases may still be problematic.

## 4.3 Cluster Average

Lastly, a cluster average was applied at each turbine. This is similar to consensus in that only measurements from the nearest X turbines are used, where X = 15 for this study. However, cluster averaging uses a simple weighted average rather than the iterative, consensus-based approach presented in Section 3.1. This uses the network topology also used for the consensus-based approach to determine the wind direction at a turbine:

$$x_i = \sum_j^{N_i} w_{ij} x_{j,measure} \tag{9}$$

where the weighted average wind direction at each turbine is determined by a subset of turbines, $N_i$. In the example in Section 6, $N_i \in \mathbb{R}^{20}$ This is similar to the weighted average method except only a subset of turbines are used rather than all of the turbines. Using only this subset may be helpful in excluding information from turbines that are far away.

## 5 Comparison of Methods: Wind Direction Estimation

This section demonstrates the benefits of the consensus approach in Section 3.1 compared to the alternative methods presented in Section 4.

## 5.1 Small Wind Farm - Fault Detection

First, the four different methods in Sections 3.1-4 were tested on a 6 turbine wind farm to demonstrate the performance when a bias or fault is experienced at a turbine. The 6 turbine wind farm is shown in Figure 1a. The network topology used for the consensus approach, Section 3.1, and the cluster averaging, Section 4.3, is shown in Figure 1b using the 3 nearest neighbors clustering strategy. The different colors in Figure 1b represent the different clusters. For example, the darker blue lines indicate that Turbines 2, 3, and 4 are communicating with Turbine 5.

The wind direction is simulated from 270° and a fault is introduced at Turbine 0 such that the wind turbine is measuring a 90° offset from the wind, i.e. a wind direction of 0°. Such an error is possible in the field for a number of reasons including a drifting yaw position signal or a bent wind vane.

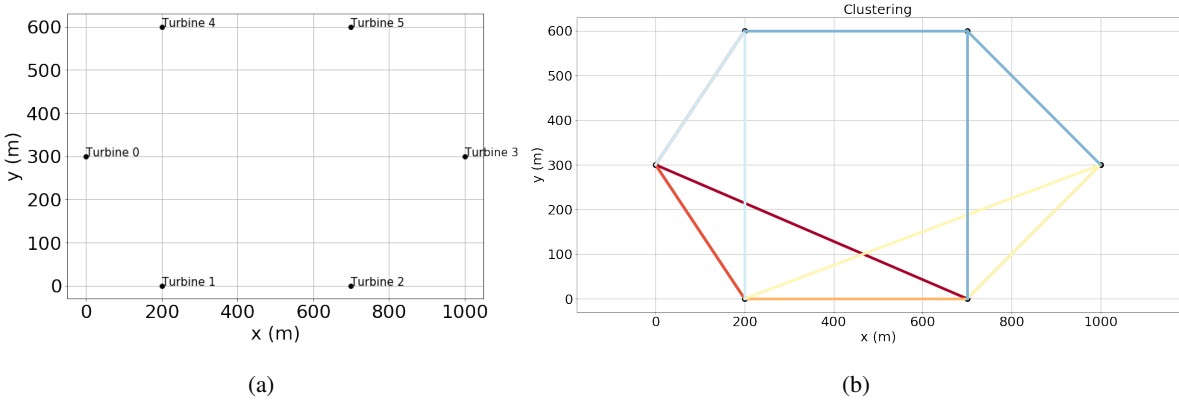

**Figure 1.** (a) Example 6-turbine wind farm. (b) Example clustering based on the nearest 3 neighbors. The different colors represent the different clusters. For example, the darker blue lines indicate that Turbines 2, 3, and 4 are communicating with Turbine 5.

Figure 2 shows the results of implementing the different methods described in Sections 3.1-4. The top left plot shows the 'measured data' vs. the true wind direction. The top right plot shows the estimated wind direction at each turbine using the consensus approach described in Section 3.1. This plot shows that, through its iterative approach, consensus is able to determine the actual wind direction at Turbine 0 as well as at all of the other turbines. In evaluating the three averaging approaches in

Section 4, the error introduced by a fault at Turbine 0 spreads at least to the nearby turbines or in the case of the simple average, across the entire wind farm. The resulting mean absolute error of each scenario is shown in a the bottom right subplot of Figure 2. The consensus algorithm is able to outperform all of the other methods in this faulty sensor scenario. The mean absolute error shows that these other methods reduce the "average" error across the wind farm. This metric is important when assessing the accuracy of a method across a wind farm as will be shown in a larger wind farm example in Section 5. However,

in this example, there is only one fault/error and the plots in Figure 2 show that the error has spread to more turbines. In this case, it is critical that the consensus algorithm is able to identify the erroneous wind direction signal and minimally impact the other turbines in the wind farm. This will have implications when implementing advanced wind farm control strategies like wake steering.

## 5.2   Large Wind Farm

Next, the different methods were applied to simulated data of a real wind farm. The wind farm is a subset of wind turbines used in the Wind Forecasting Improvement Project, also known as WFIP2 (Pearse et al. (2017)). A subset of those turbines are shown in Figure 4a, resulting in more than 100 turbines. Note, the $x$ and $y$ axis scale has been removed for proprietary reasons. The wind farm includes two met towers with sensors at 50 m and 80 m elevation and a sodar. SCADA data was collected at 1-minute time intervals from individual turbines over approximately 8 months. The data channels of interest were

the perceived wind direction (yaw position combined with the yaw error), wind speed, and measured power at each turbine. The latitude and longitude values of each turbine were transformed into Universal Transverse Mercator (UTM) coordinates to

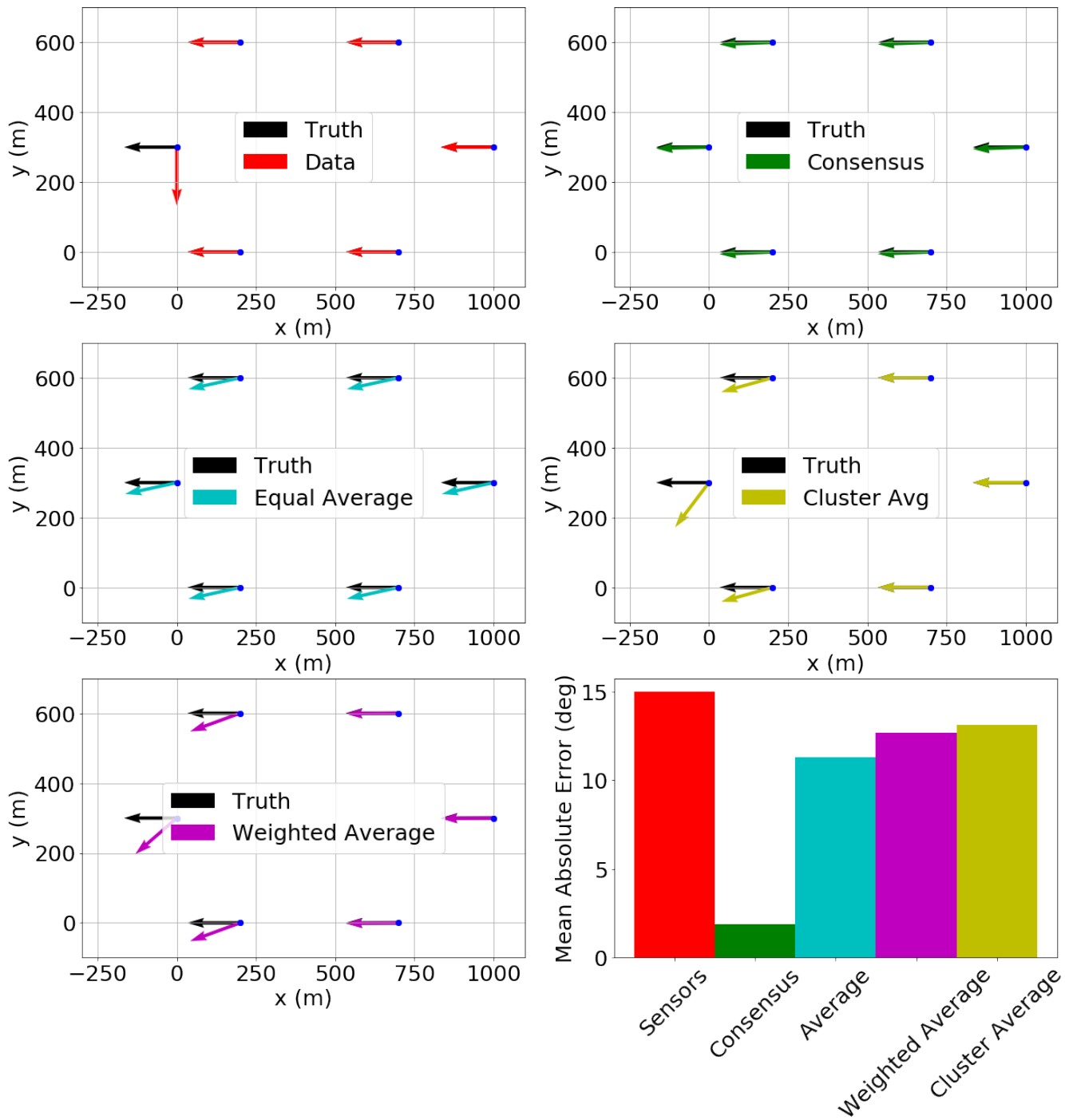

**Figure 2.** This 6 turbine example demonstrates the effectiveness of the different methods in the presence of a fault in the Turbine 0 wind direction sensor. The top left plot shows the 'true' data and the 'measured' data. Note, the arrows are pointed in the upwind direction. The top right plot shows the results of the consensus algorithm. The middle left plot shows the results of equal averaging across the wind farm. The middle right plot shows the cluster averaging approach and the bottom left plot shows the weighted averaging approach across the full wind farm. The mean absolute errors of each of the methods are shown in the bottom right.

provide approximate distances in meters between turbines. In addition, data was available for the same time period from the met towers and the sodar. The met towers had data at 1-minute time intervals and the sodar had data available at 10-minute intervals.

To assess the performance of each method, the wind direction data was simulated to provide a set of 'truth' data. The
simulated wind direction was generated based on the change in wind direction seen across the wind farm with added noise. First, the change in wind direction was observed in measurements from met towers in geographically different locations within the farm. Next, noise was added on top of this change in wind direction across the wind farm. Using the met towers, there was an average of $30°$ wind direction change across the wind farm. A standard deviation of $10°$ was added on top of the wind direction change to simulate noisy wind direction signals recorded at each turbine. Figure 3a shows the 'true' underlying wind
data based on the $30°$ wind direction change across the wind farm and Figure 3b shows the 'measured' wind direction with noise added on top of the 'truth' data. Note, the 'measured' data in Figure 3b visually looks similar to the actual SCADA data recorded, see Figure 7a.

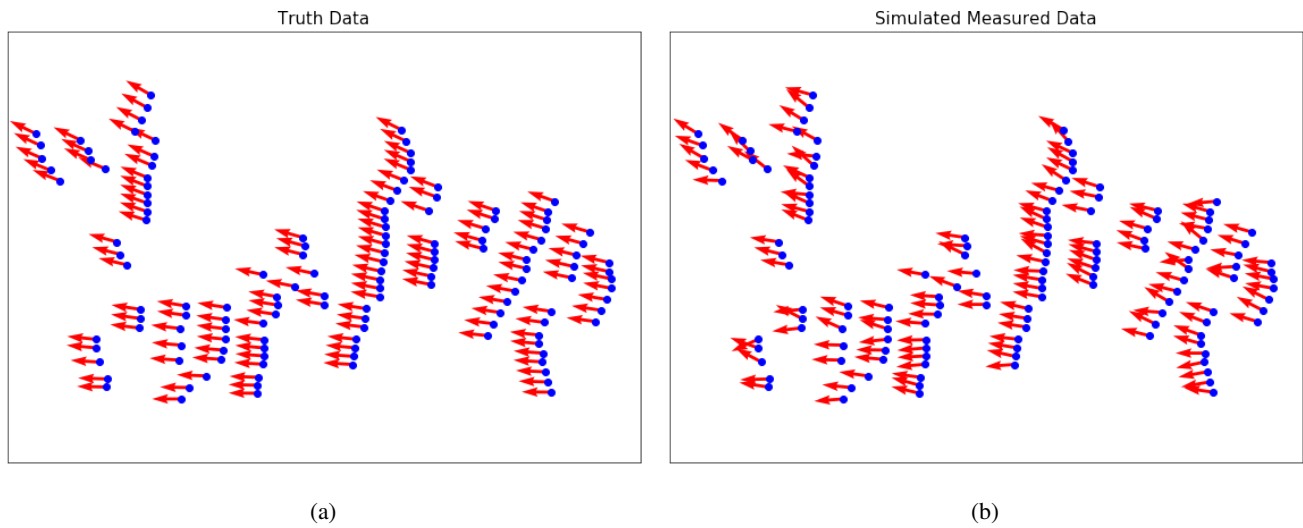

(a)                                                                                       (b)

**Figure 3.** (a) This plot shows the simulated 'true' underlying wind direction data based on the $30°$ wind direction change across the wind farm. (b) This plot shows the simulated 'measured' wind direction with noise added on top of the 'truth' data. Note, the arrows are pointed in the upwind direction.

### 5.2.1  Network Topology and Sensitivity Analysis

Using the simulated SCADA data, the network topology and the penalty parameter, $\lambda$ from (4), could be determined using a
sensitivity analysis. The results are shown in Figure 5. First, the optimal penalty parameter $\lambda$ was determined to be $\lambda = 60$ based on Figure 5a. Figure 5a shows that a small or large $\lambda$ produces a larger error. A small $\lambda$ corresponds to little consensus among the turbines and a large $\lambda$ encourages total consensus among turbines. Figure 5b shows the sensitivity of the mean

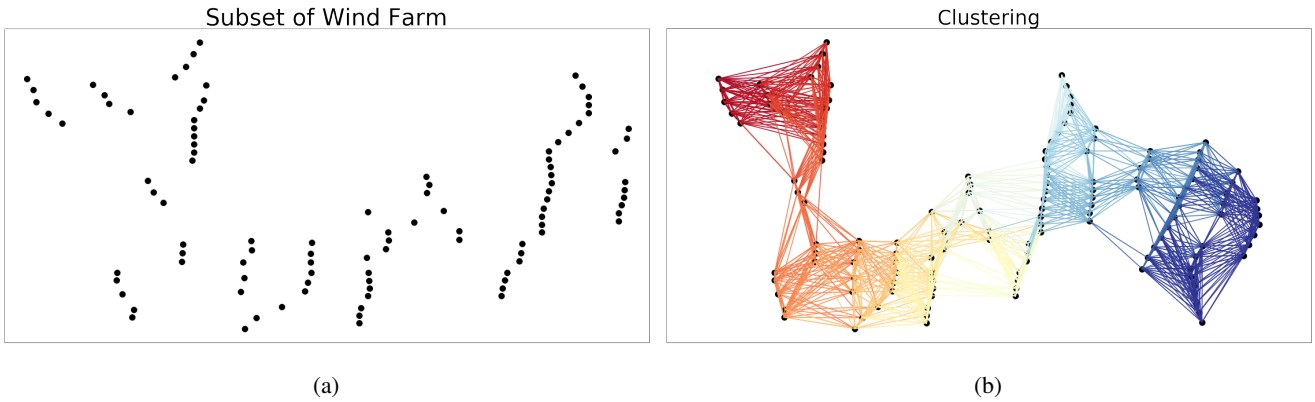

(a)                                                        (b)

**Figure 4.** (a) This figure shows a subset of the turbines used in this study. (b) Connecting undirected edges indicating all groupings within the wind farm based on distance from each wind turbine to its neighbors.

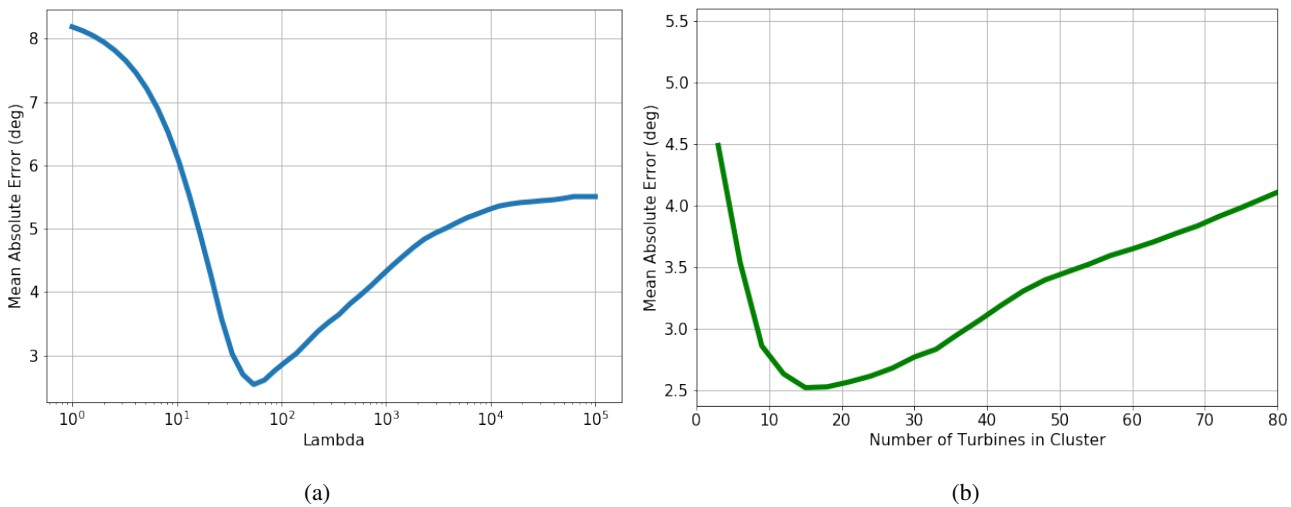

(a)                                                        (b)

**Figure 5.** (a) This subplot shows the mean absolute error computed with consensus using a range of penalty parameters, $\lambda$. (b) The cluster size was varied and resulting mean absolute error calculated to understand the best number of turbines to communicate with.

| Method | Mean Absolute Error | Maximum Absolute Error |
|---|---|---|
| Consensus | 2.68° | 9.54° |
| Equal Averaging | 6.21° | 16.01° |
| Weighted Averaging | 3.87° | 13.39° |
| Cluster Averaging | 3.45° | 12.10° |
| Sensors | 8.25° | 30.01° |

**Table 1.** Results of Different Methods. The mean absolute error column is also plotted in the lower right subplot of Figure 6. The consensus method also has the smallest maximum error.

absolute error to the clustering size used by the consensus algorithm. This subplot shows that communicating with only a few nearby turbines produces large local errors, since an individual turbine does not receive enough additional information to make an informed decision on the actual wind direction. In addition, communicating with too many turbines also produces large errors. If a turbine communicates with too many other turbines, all the turbines end up agreeing on one wind direction across the farm. For this wind farm simulated data set, the consensus approach provides the best results with approximately 15 turbines. Therefore, the remainder of the results are shown with turbines communicating with the nearest 15 turbines, see Figure 4b.

### 5.2.2 Comparing Different Methods

Next, the different methods described in Section 3.1-4 were compared across the large wind farm. Figure 6 shows the results of the different methods applied to this large wind farm with a wind direction change of 30° and a standard deviation of 10°. The results indicate that the best results are achieved with the consensus algorithm, but the other methods perform similarly and are not as computationally expensive. They can reduce the error at some individual turbines, especially when no faults or additional biases are present. If corrected signals are included where no additional biases are present, the cluster averaging approach would be a good computationally efficient option. The total error results of each of the methods are summarized in Figure 6 in the bottom right plot. In addition to the mean absolute error, the largest absolute errors were also recorded for each method and those results are shown in Table 1.

### 6 SCADA Data Analysis Using the Consensus Algorithm for Power Analysis

In Section 4, we showed that the consensus algorithm is better able to estimate wind direction than the equal average, weighted average, and cluster average methods. In this section, we examine opportunities enabled by the algorithm, including with respect to power estimation and power curve calculation. First, consensus can capture the varying wind direction across the wind farm. Next, it can predict power loss by calculating misalignment in the turbine.

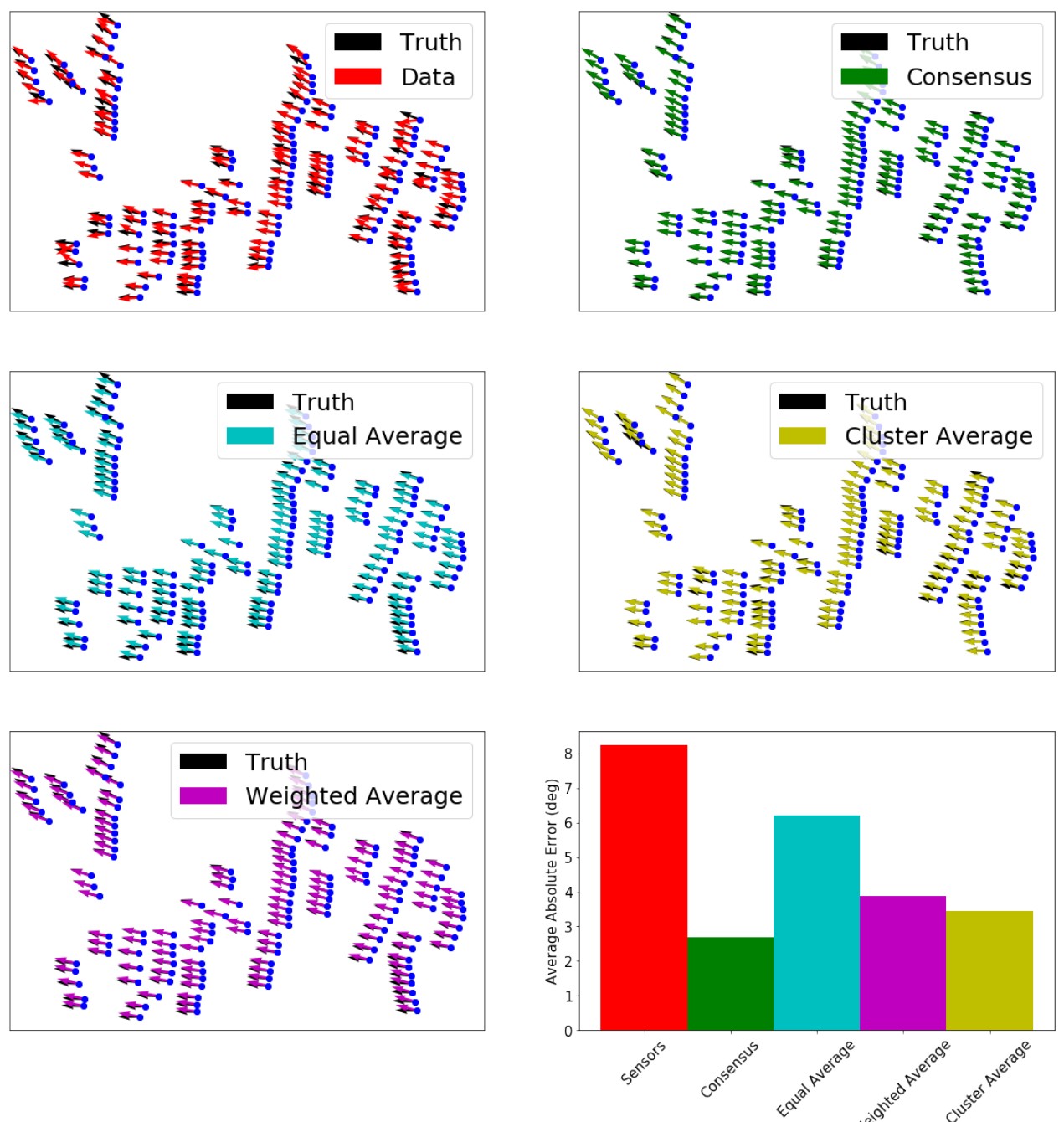

**Figure 6.** This figure demonstrates the performance of the different methods on a larger wind farm than the one shown in Figure 2 with noisy signals and a wind direction change across the wind farm. The top left plot shows the 'true' data and the actual data. The top right plot shows the results of the consensus algorithm. The middle left plot shows the results of implementing equal averaging across the wind farm. The middle right plot shows the cluster averaging approach and the bottom left plot shows the weighted averaging approach across the full wind farm. The mean absolute errors of each of the methods are shown in the bottom right subplot.

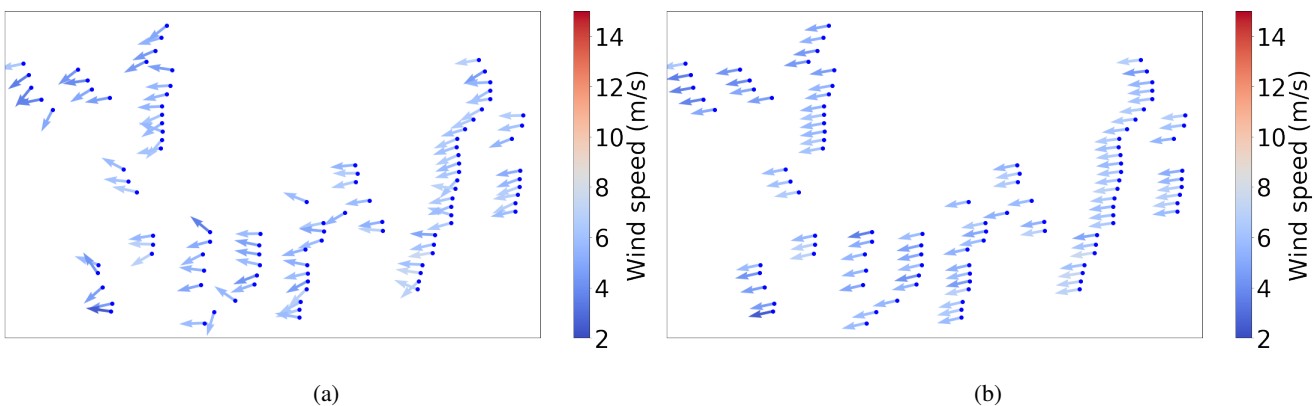

**Figure 7.** (a) The wind direction recorded at each turbine across the wind farm at one time step using actual SCADA data recorded at each turbine. (b) The wind direction estimated by the consensus algorithm at the same time step.

The consensus algorithm was used to post-process SCADA data to understand the possible gains of this approach. First, the wind direction was examined at one time step with and without the consensus algorithm. Figure 7a shows the wind direction recorded at each wind turbine for one time step. This subplot shows the variability across the wind farm and the disagreement among turbines. Figure 7b shows the wind direction determined from the consensus algorithm at the same time step. Each time step takes 0.5 s to compute, in serial on a desktop computer, using the algorithm described in Section 3.1. The output of the consensus algorithm shows a smoothly varying wind direction across the wind farm. One implication of smoothly varying wind direction is that it may may reduce the yaw motion of the yaw controller and the yaw drive in that turbines are not chasing local wind gusts that only last for a short time. Figure 8 shows the terrain and the corresponding color-coded wind direction and indicates that the wind direction varies with terrain, an effect that this algorithm is able to capture even in complex terrain. In particular, a strong change in wind direction is detected near the canyon in the north-central part of the wind farm.

Next, the results of the consensus algorithm were used to determine the wind direction at the location of the sodar on the outside of the wind farm by interpolation from the individual turbines, i.e. the wind direction was determined at each turbine and the results were interpolated to the sodar location within the wind farm. The results were compared with the time series data recorded by the sodar as shown in Figure 9. The top plot shows the time-series wind direction measured by the sodar in blue and the estimate based on the consensus algorithm in red. This figure shows good agreement between the estimated wind direction and the wind direction measured by the sodar. The 100 hours shown in Figure 9 were chosen to demonstrate the performance of the algorithm under large wind direction changes due to meteorological drivers. Figure 9 (bottom) shows the error between the estimated and actual signal recorded by the sodar. The points are color-coded with respect to wind speed. The largest errors are experienced at low wind speeds, typically at or below cut-in. Figure 10 shows the error distribution of the consensus algorithm compared with the sodar for wind speeds greater than 4.0 m/s.

Next, to demonstrate the benefits of the estimated wind direction at each turbine, the estimated wind direction was used to determine the error, between the wind direction signal and the consensus-based wind direction signal, experienced by each

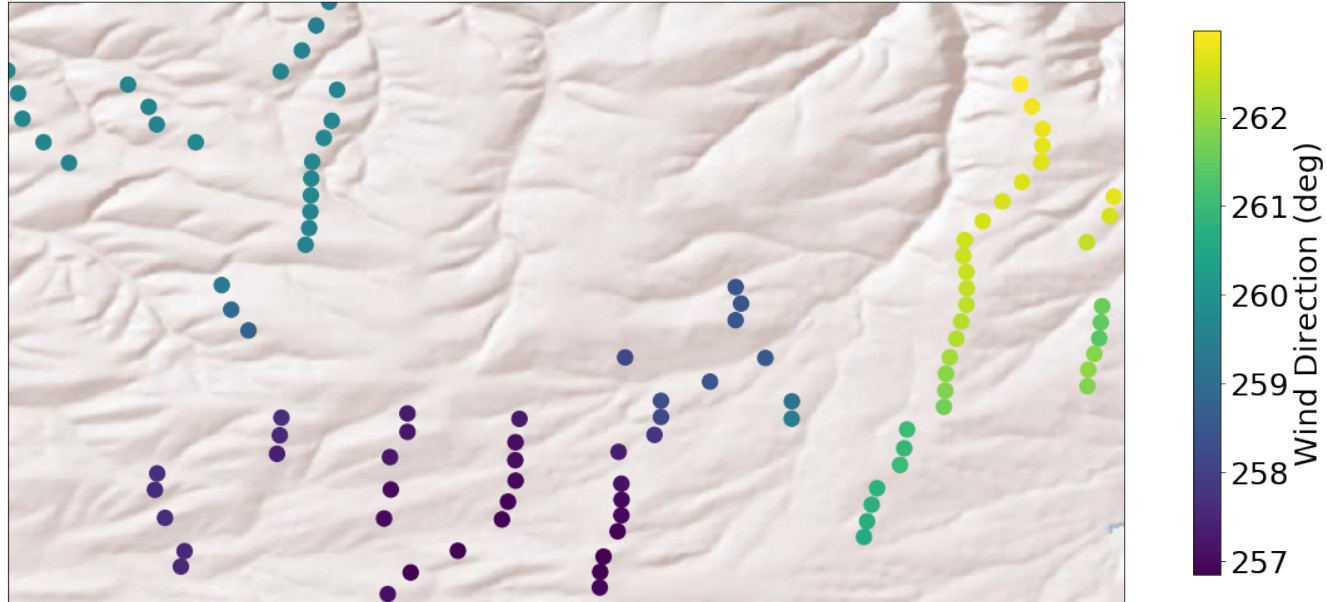

**Figure 8.** The consensus-based wind direction is shown across part of the wind farm based on the color of each turbine with the terrain plotted in the background. This shows the effects of terrain on the wind direction and indicates that the wind direction can vary across the wind farm.

turbine across the wind farm. The error was calculated between the estimated wind direction and the measured wind direction at the turbine. Figure 11a shows the power curve, computed with 95% confidence intervals, of one turbine with small wind direction errors (less than $1°$) in green and large wind direction errors (greater than $10°$) in magenta. Using the estimated wind direction, we can identify when turbines are operating in misaligned conditions and correct dynamic yaw misalignment in the
yaw controller.

    Next, the average power loss of a turbine was computed for different amounts of wind direction error using a wind direction error of less than $1°$ as the baseline from the SCADA data. Figure 11b shows the results of the average data, across all turbines across 500 hours, in black. The percent of the 500 hours of data used to compute the power loss at each offset is shown; i.e., 48.5% of the data has less than $5°$ offset. This plot indicates that some turbines across the wind farm could
be spending a significant amount of time misaligned. In literature, turbines operating in yaw misalignment have a loss of power that is proportional to $\cos(\theta)^{p_P}$ (Gebraad et al. (2016)), where $\theta$ is the misalignment angle and $p_P$ is determined empirically (Fleming et al. (2017)). The value of $p_P$ has been shown to be between 1.0 and 2.0 (Gebraad et al. (2016); Fleming et al. (2017)). In Figure 11b, the SCADA data shows the percent decrease in power vs. misalignment, where misalignment is calculated compared to wind direction consensus, most closely follows a $\cos(\theta)^{1.4}$ relationship. The loss in power is consistent
with literature, which again indicates that the consensus-algorithm-estimated wind direction is a reasonable estimate of the wind direction at each turbine. Having a better wind direction measurement for the yaw controller of a turbine could improve

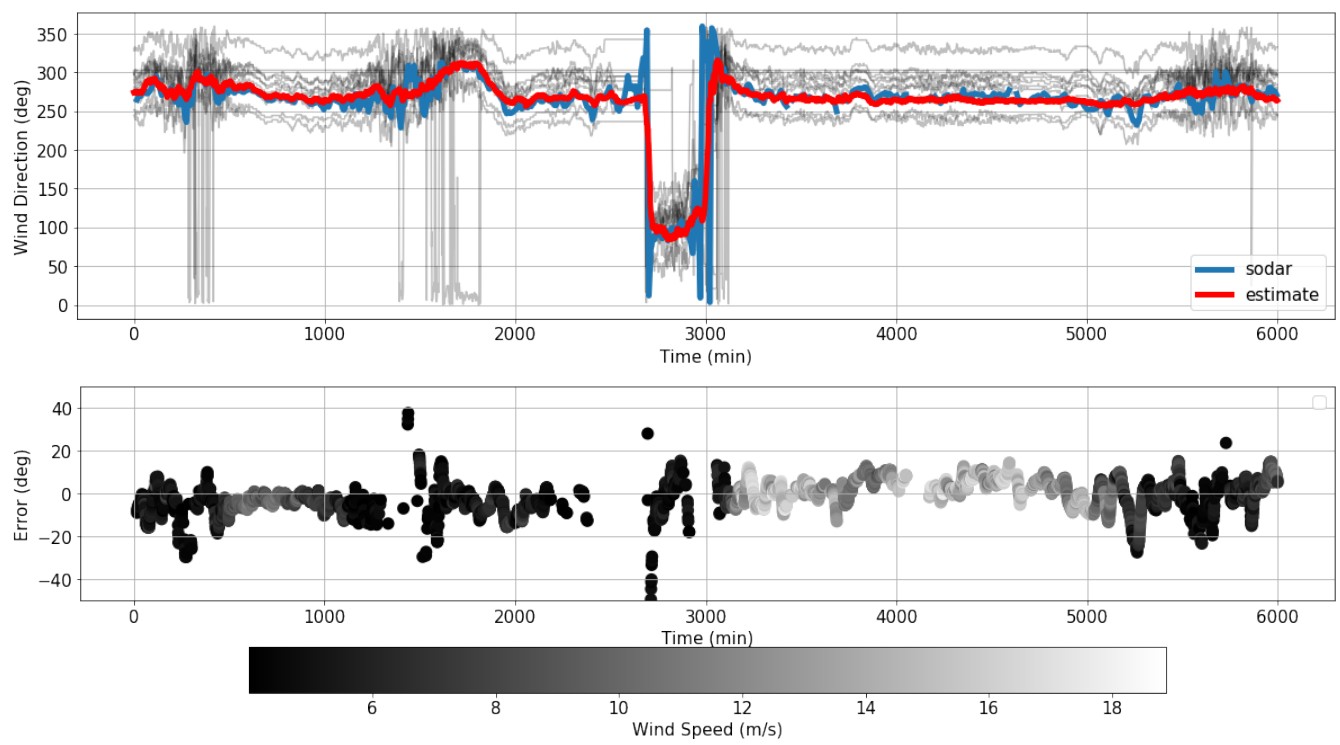

**Figure 9.** (Top) Comparison between the estimated wind direction calculated via consensus at the location of the sodar (red) and the actual wind direction recorded by the sodar (blue). The wind directions recorded by the nearest 10 turbines are shown in gray. (Bottom) Error between the estimate and the actual wind direction recorded at the sodar. The dots are color-coded based on wind speed.

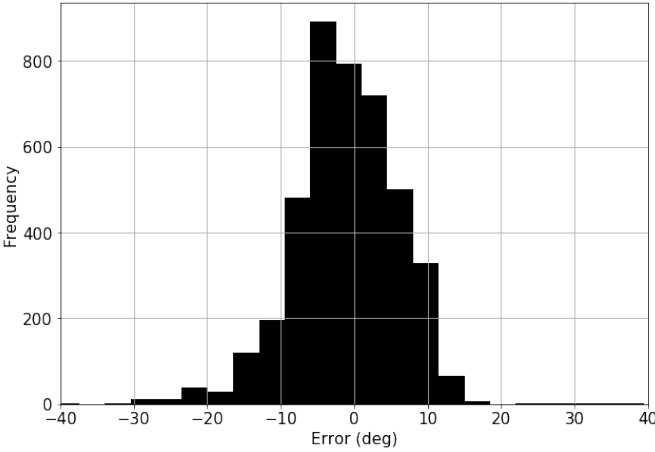

**Figure 10.** Error distribution of the estimated wind direction compared with measurements from the sodar. Note the measurements of the sodar are interpolated to match the 1 minute time series from the consensus algorithm.

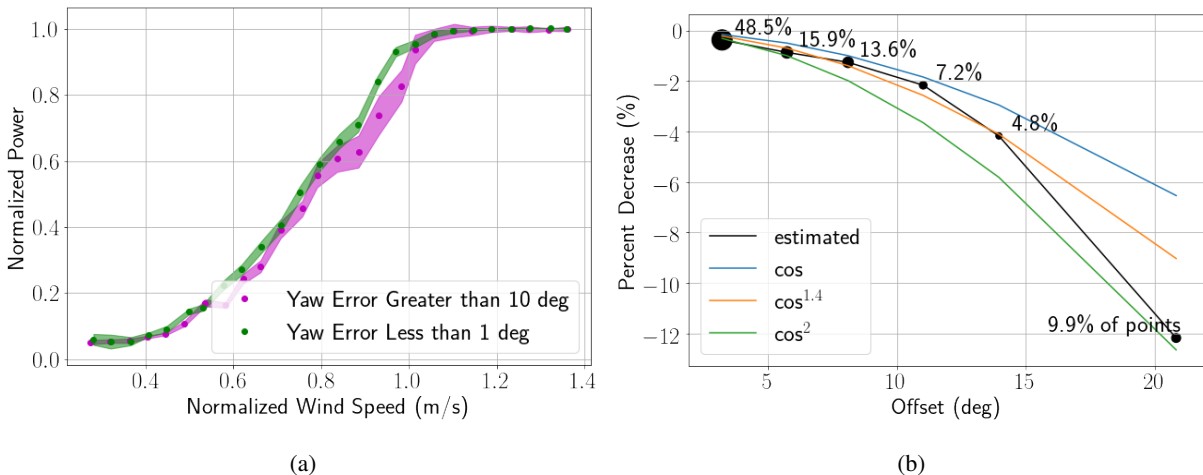

**Figure 11.** (a) The power curve of a single turbine was computed using 0.5 m/s bins over 500 hours of data. The power curve is plotted for small wind direction errors of less than 10° (green) and large wind direction errors of more than 10° (magenta). (b) The power loss is computed for different yaw offset angles. The trend is consistent with cosine power laws seen in literature.

yaw misalignments and reduce the amount of yawing a turbine performs and might improve the effectiveness of wake steering (Fleming et al. (2014a)).

Finally, one additional metric was used for assessing the potential value of the consensus algorithm. We calculated the relative power performance of the turbines as they experienced large (>20°) and small (<10°) wind direction errors compared to two baselines: (1) the sodar in the wind farm, and (2) the estimated wind direction from the consensus algorithm. In this analysis, we first eliminated erroneous data such as power values more than 13% over rated power and less than 0. We used 10-min sodar data and 1-min turbine power and nacelle position data and calculated the statistics including the mean, median, and standard deviation of the power at each individual turbine based on wind speed bins of 1 m/s. An average power curve was computed for the wind turbines in this wind farm after removing further outliers, defined as data points that were outside of two standard deviations above or below the median in each 1 m/s wind speed bin. Based on a separate analysis using a power curve analysis for each individual turbine, we determined that some turbines likely experienced drift in their yaw-position sensors, causing it to appear that they had regularly large wind direction errors despite being oriented correctly into the wind. We therefore removed data from turbines with consistently high, inexplicable wind direction errors compared to the consensus algorithm or the sodar.

Figure 12 shows results from this second power curve analysis that attempts to determine the effects of wind direction error on the power of a turbine. The wind direction error was computed using a sodar on-site and the wind direction estimated with the proposed consensus algorithm. The top-left plot shows the difference in the power-curve analysis when a large versus small error is detected using the wind direction reference from sodar measurements. The bottom-left plot shows the percent difference between these two power curves at each bin, where most percent differences are less than 1%. In other words, the

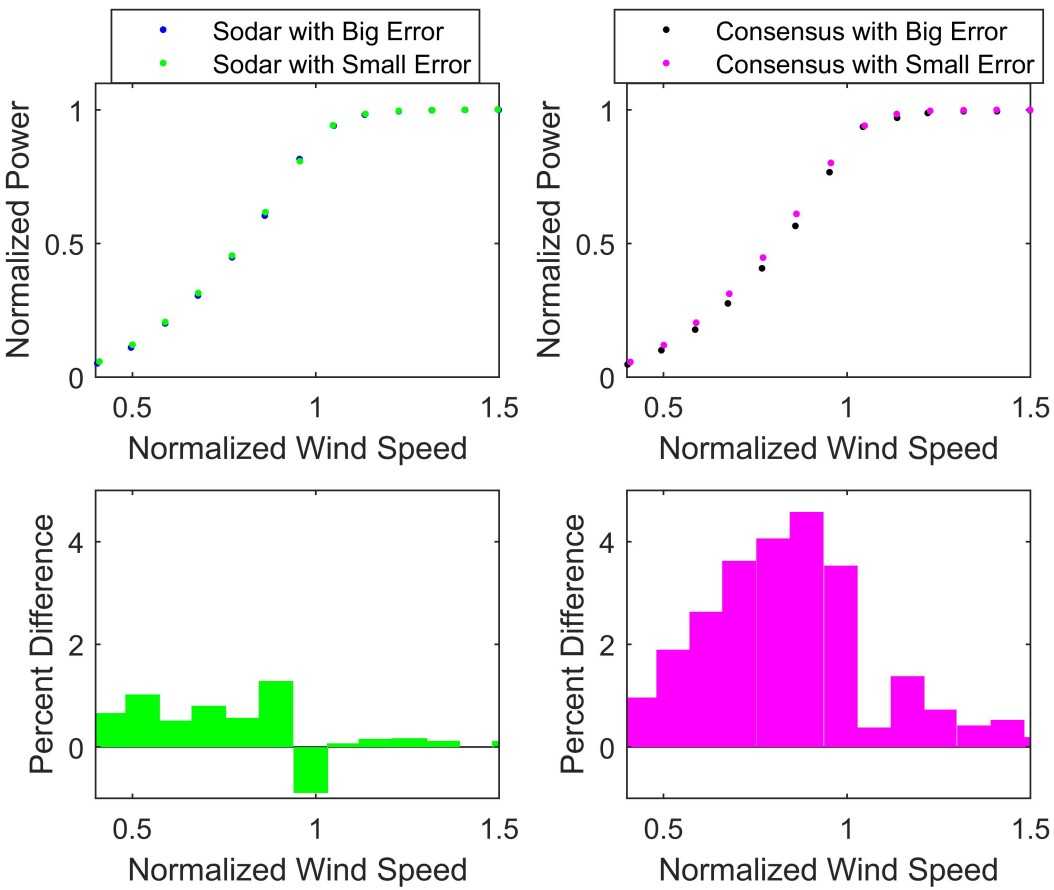

**Figure 12.** (top row) Power curves based on binned data for 1 m/s wind speed bins across all turbines, and (bottom row) percent difference between binned power from small wind direction errors to large wind direction errors. Positive percent difference indicates more power is generated when wind direction errors are small. The left column is based on comparisons between turbine wind vanes and the sodar and the right column is based on comparisons between the turbine wind vanes and the consensus algorithm.

sodar reference is almost the same as the turbine sensor reference. The top-right plot shows the difference between the two power curves when using the consensus-based wind direction estimate as the wind direction reference. The bottom-right plot shows the percent difference between the small and large wind direction errors detected with the wind direction estimate, which shows much larger differences than when using the sodar as wind direction reference. The consensus algorithm, therefore, appears better able to detect true wind direction errors than the sodar.

The implications of the results presented in Figures 11 and 12 suggest that with a modified wind direction signal, it might be possible to account for these wind direction errors in real time and improve the performance of an individual turbine by taking advantage of consensus-derived wind direction estimates from additional data available within the wind farm. Many data-analysis factors impact the quantitative performance, so we do not assert any specific quantitative gains for the consensus algorithm compared to turbine wind vanes or sodar, but instead point to some qualitative differences to motivate future research. Given the data-processing decisions explained in the Figure 12 description, it appears that the consensus algorithm more closely predicts the actual wind direction error across turbines than does the sodar in that small errors measured with consensus result in higher mean powers for wind speed bins below rated. This result is expected because there is only one sodar in the wind farm and a significant amount of spatial variation in the wind direction, which the consensus algorithm is able to capture. This analysis indicates that by using only SCADA data in the way outlined in this article, it is possible to detect dynamic yaw misalignment. A corrected wind direction input based on this algorithm could be used with the yaw controller, which might be able to minimize yaw misaligned conditions. Lidars have been used to date to correct for yaw misalignment. However, due to the limitations in scanning distances, lidars have only been able to correct static misalignment. This approach allows for more reliable wind direction measurements that correspond to larger time and space scales, which can ride through local wind variations with small timescales and might avoid yawing prematurely.

## 7   Conclusions and Future Work

This article presents a framework for a cooperative wind farm where turbines benefit from increased communication, i.e., sharing of data. The specific example that has been presented here is a wind direction consensus algorithm that uses information from nearby turbines to determine the wind direction. The results indicate that this strategy is able to detect wind direction errors and is supported using a power curve analysis and expected power loss from misalignment from the literature. By increasing communication between turbines, it is possible to improve the performance of turbines in a wind farm. The proposed consensus-based approach was also compared with averaging methods and outperformed these methods especially when handling turbines with faults and/or biases in the data.

Future work will improve and extend the consensus approach beyond wind direction estimation alone. An early step will determine the optimal number of connections between turbines given the layout and terrain features as well as allowable computation time. Next, the network topology chosen could facilitate short-term forecasting in a wind farm. For example, it takes minutes for wind to propagate downstream. Turbines that exist upstream could communicate to connected downstream turbines the near-term conditions including wind direction changes that could mitigate extreme loading events. In addition, the

consensus approach will investigate the sensitivity to various turbulence levels in the atmosphere. This could be incorporated into the consensus algorithm itself in the form of a stochastic optimization.

Finally, future work will include incorporating wind speed data to have a better estimate of the atmospheric conditions at each turbine. In addition, future work can use this framework for fault detection in wind direction sensors. Sensors that stray far from the consensus can be flagged for maintenance and if wind direction sensors are offline, information from nearby turbines can be used to continue operating the turbine until maintenance can be scheduled and completed. In summary, this framework for cooperative consensus lays the groundwork for many different avenues for cooperative, autonomous wind farms.

*Acknowledgements.*

This work was authored by the National Renewable Energy Laboratory, operated by the Alliance for Sustainable Energy, LLC, for the U.S. Department of Energy under Contract No. DE-AC36-08GO28308. Funding provided by the U.S. Department of Energy Office of Energy Efficiency and Renewable Energy, Wind Energy Technologies Office. The views expressed in the article do not necessarily represent the views of the U.S. Department of Energy or the U.S. Government. The U.S. Government retains, and the publisher, by accepting the article for publication, acknowledges that the U.S. Government retains a nonexclusive, paid-up, irrevocable, worldwide license to publish or reproduce the published form of this work, or allow others to do so, for U.S. Government purposes.

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
