# Peer review of "Wind Direction Estimation Using SCADA Data with Consensus-Based Optimization"

_Wind Energy Science, 2018_

## Referee Comment (RC1) · Anonymous Referee #1 · 29 Oct 2018

This paper is well-written and well-structured. The presented experimental data is interesting. However, I have my doubts about the usefulness of the proposed method. In the remainder of this review I will further elaborate on this issue.

Major comments: - The authors talk, in the introduction of the paper, about the robustness of the algorithm (..robustly estimate ..., robustly calculating...). It is not clear how the authors define robustness and there is no proof of robustness (either in terms of eq.'s or simulations). The solution of eq.2 is trivial. I believe it is good to make assumptions regarding the measurements (variance or potential bias). - Main criticism: The main contribution of this paper is to add (3) to (2). The authors decide to keep the weights $w_i$ constant. The authors basically take the measurement of the neighbouring into account in a rather ad-hoc way. I have the following questions: o Why should this

work? o What is the effect of w's on the results? o The most trivial solution is to use a spatial filter. Why not creating a simple spatial filter. (there are different extremely simple implementations possible 1. average of all turbines 2. average of a cluster and you distance to central point as weight). These trivial solutions need to be explored and compared with the proposed method. o Related to my previous point, isn't it a bit of overkill to use the proposed machinery - Section 3.2 can be skipped. Just state that you use ADMM to solve this problem - I would also suggest to include a simulation study in which the sensitivity of the proposed method is explored.

Minor comments: - In the abstract the authors should add more details regarding the methodology they use - Pg 2, line 5, I believe that modern wind turbines also employ estimators to get an estimate of the wind speed and wind direction - Section 2 can be shortened. The information density is rather low - Pg 7, rho is not defined in (4)-(5) but suddenly it appears in the experimental section. I believe it is a tuning variable of the ADMM algorithm and I don't understand why it should be tuned for the experiment - Check consistency of the literature list.

---

## Referee Comment (RC2) · Anonymous Referee #2 · 20 Nov 2018

This paper proposes an algorithm to estimate the wind direction using a consensus algorithm within relatively small clusters of turbines (∼10 turbines) in a wind farm. The proposed algorithm is based on available SCADA measurements of wind direction at the turbine level. The proposed algorithm does not rely of a physics-based model of the turbines or the flow field.

The use of consensus algorithm for this application is interesting. The presentation obscures the contribution. There are high-level issues that start with the title and continue with the presentation of certain material. There are technical questions to be addressed. Lastly, there are minor issues that would need to be considered by the authors.

High-level issues:

-The title does not seemed focused on the problem considered. That is, "wind direction estimation using SCADA data" is the topic of the paper. Thus, why "mention autonomous wind farms? In fact, what do the authors mean by an autonomous wind farm?

-One might guess that the answer is in section 2, but this section is not that relevant to the paper; an autonomous wind farm is loosely defined as one that "self-organizes into groups, monitors, and controls its performance in real time based on existing SCADA data." But this paper has no self-organization and no control.

-Section 2 then goes onto discussing directed and undirected networks. What is the point of discussing directed networks when only undirected ones are used later? This distracts from the main point of the paper. Another point of distraction is the title of section 2.3; there is no control in this paper.

Technical issues

-Elaborate on the selection of rho and lambda. Are these parameters the same for all 10-turbine clusters in the farm?

-How do you use the SCADA data to "interpolate" wind direction at the two met tower locations?

-In section 5 you validate the approach at the sodar location (fig. 5). This is a good idea and it would be helpful to know the sodar location.

-After comparison with the sodar, there is comparison between estimated wind direction and SCADA measured direction. You use consensus amongst 10 turbines to mitigate errors at the single turbine SCADA measurement. Thus, this is confusing to me. Can you clarify? Can you present the uncertainty in the SCADA data? How does the result of the consensus algorithm compare with averaging the wind directions obtained from some of the turbines in a cluster (which is simpler)?

-It would help to review the literature from the meteorology community on the topic of

wind direction estimation from multiple sensors.

Minor issues:

-Equation (8), check the variable being optimized.

-What do you mean by x being non-convex after equation (11)?

-Where are the sodar and met towers in Fig. 2(a)? (4th sentence in paragraph one of section 4.)

---

## Referee Comment (RC3) · Anonymous Referee #3 · 23 Nov 2018

This approach is an interesting alternative to make use of the information from the neighbouring turbines to correct the wind direction (WD) signal from SCADA at the turbine locations. Although it promises valuable contribution to the often overlooked, messy WD data processing, it could benefit from a more thorough investigation to study the sensitivity of the developed consensus to some of the local inflow characteristics.

More detailed comments/questions are listed down below:

**p.1 : line 9-10 (Abstract) – "Oftentimes, measurements made at an individual turbine are noisy and unreliable" is rather a blanket statement and it not clear if only the WD signal is referred to or not. It should be noted that SCADA does/might provide some crucial information, and used in many applications in WF operation, so a re-wording is suggested. # p.1 : line 11 (Abstract) – By taking the nearby turbines into account, the**

[Figure]

WD signal at an individual turbine can be improved, however, it is equally important to not to lose the local information in certain applications. This remark will be repeated several times in these comments.

**p.2 : line 3-4 – What is actually meant by the unnecessary yaw movements? This should be elaborated further... if it is the turbine responding "too fast" to the WD changes, then it can argued that a simple low-pass filter might be enough. However, in the field it is generally the opposite effect as the turbine is generally "too slow" to respond to the highly variant WD (due to its inertia) which is one of the factors that might affect the behaviour of the power curve for highly turbulent flows for example.**

**p.3 : line 1-2 – "...facilitate wake steering WFC..." and more - the benefit of having a reliable WD signal is important for any kind of wake modelling really, including other WFC scenarios, operations management and conditions monitoring.**

**p.3 : line 18-19 – Is the "proximity" to define the connected turbines estimated in a relative proximity manner, i.e. taking the (reference) incoming WD into account? Otherwise, especially for the investigated terrain, it might be a risk to connect the flow-wise uncorrelated turbines, e.g. highly different turbulence levels (hence very different variance in the local WD), etc.**

**p.4 : line 6-7 – It is very true that the turbines that are several kilometers apart would experience highly different local WDs, however, it could also be the case when the turbines are much nearer. It should be clarified how near is nearby (possibly depending on local flow conditions) and how much of the local characteristics are kept and how much of it is smeared among a larger area.**

**p.5 : line 13-14 – "...based on nearest 10 turbines" Is there any particular reason for the selection? As indicated in the previous comment, how much of the local information is intended to be kept in such a network should be indicated and the reasons should be argued. (e.g.Figure 2b shows some of the connected turbines are much further apart than the others in the same local network. It is hard to argue that the contribution**

from those turbines should be included - especially with the same weight, as will be mentioned later).

**p.6 : line 6-7 – "The objective function, fi(xi)..." sentence should be omitted as it is confusing compared to eqn. (2)**

**p.7 : line 2-3 – Why the weight is equal to 1? Especially given that the correlation between some of the turbines would be lower than others, simply due to distance and local terrain differences.**

**p.8 : line 24-25 – How is lambda and rho related to/different than the weight wjk? Their explicit definition and tuning procedure should be included.**

**Overall Section 3: – It should be discussed as to why this rather complex methodology is/should be selected over an educated but simple interpolation (maybe combined with a low pass filter if the noise is a serious concern).**

**page 9 : Figure – The met towers should be visible in Figure 2 to understand the validation cases**

**page 10 : line 4 – Is the time step shown in Figure 3 different from the 500h used during tuning the parameters lambda and rho (i.e. is it an 'independent' dataset?) It is hard to assess how much of a difference is to be expected between the neighboring turbines, given the terrain (applies also for Figure 4). In that sense, the visual comparison might be, at least partially, misleading.**

**page 10 : line 16-17 – The error sensitivity to wind speed (WS) might be due to the turbulence intensity (TI) behavior with respect to WS. Another color coding or, in general, an additional sensitivity analysis to TI at the sodar would tell a more detailed story.**

**page 11 : Figure 4 – Is it also a "snapshot" or an average over some period? Again, one could argue that in 'real-time' we would/might expect higher variance over a relatively big terrain.**

**page 11 : line 2 – Again, due to the turbine's inertia, we generally see the opposite effect in fact - turbine is having a hard time to catch up with (truly, not just noisy) variable WD sometimes. Therefore, it is generally expected the (non-intentional) misalignment to occur due to the bias in the sensors, rather than the noise.**

**page 11 : line 3-4 – An important factor in terms of defining the true error might be the difference between the wind vane measurements and the actual nacelle position (as mentioned very briefly later in the paper during the filtering of the data for the last validation case). In most of the turbines' SCADA, there exist a separate signal (than the wind vane measurements, generally called wind direction) which is called Nacelle position or Nacelle direction. Those two signals differ quite a bit, especially in high turbulence cases (turbine not following the highly variant WD as mentioned earlier here). It is important to clarify what is 'corrected' in this study is only the wind vane measurements which may differ from where the turbine is actually facing.**

**page 13 : Figure 6 – The effect of the changing TI levels between the two yaw error cases defined in Figure 6(a) should be clarified.**

**page 13 : line 3-6 – Again, that might be due to the difference between the WD measurements on top of the turbines vs. Nacelle position**

**page 13 : line 13 – Suggest rewording to "truth" : reference, baseline, true value**

**page 13 : line 13-14 – Not clear how and why the error estimates from the consensus are more reliable than the sodar measurements? Depending on the equipment itself surely, sodar has been shown to agree ver well with the met mast measurements on the site (e.g. Steven Lang and Eamon McKeogh (2011) LIDAR and SODAR Measurements of Wind Speed and Direction in Upland Terrain for Wind Energy Purposes, Remote Sensing)**

**page 14 : Figure 7 – The difference between the consensus and the wind vanes seem to increase with increasing "immediate wake" effects on top of the nacelle, again**

pointing towards the (added) turbulence sensitivity. Overall, the effect of turbine not being able to follow the highly fluctuating WD, hence dynamic misalignment and under-performance, is more of a physical phenomenon due to turbine inertia. Correcting the signal would not necessarily solve that problem.

**page 15 : line 7-9 – 'Smearing' the small timescales in local WD might be useful for many other analysis, however, if the turbine avoids yawing as a result, we might risk to lose power still. Correcting the bias in the signals on the other hand, is a much more useful outcome of such an approach.**

---

## Author Comment (AC1) · 4 Feb 2019

**Review Comments for "A Framework for Autonomous Wind Farms: Wind Direction Consensus"**

Corresponding author: Jennifer Annoni

February 4, 2019

**Abstract**

The authors would like to thank the reviewer for their comments. The authors believe that this paper is much improved by addressing the reviewer's comments.

**1 Reviewer 1:**

The paper is well-written and well-structured. The presented experimental data is interesting. However, I have my doubts about the usefulness of the proposed method. In the remainder of this review, I will further elaborate this issue.

MAJOR COMMENTS:

- The authors talk, in the introduction of the paper, about the robustness of the algorithm (...robustly estimate ..., robustly calculating...). It is not clear how the authors define robustness and there is no proof of robustness (either in terms of the equations or simulations).

  *The authors agree that there is no formal robustness metric presented in the paper. This term is removed in favor of more descriptive terms such as 'reliable'. In addition, this method also automatically identifies outliers in the data. The outliers are still used in the optimization and are determined through the same consensus approach. This was not previously documented in the proposed algorithm and is now mentioned in Section 3.*

- The solution of equation 2 is trivial.

  *The authors have removed this equation.*

- I believe it is good to make assumptions regarding the measurement (variance or potential bias).

  *The variance and change in wind direction are estimated across the wind farm and is used to evaluate different approaches to estimating wind direction as well as tune the consensus-based optimization. See Section 5.2*

- The main contribution of this paper is to add (3) to (2). The authors decide to keep the weights $w_i$ constant. The authors basically take the measurements of the neighboring into account in a rather ad-hoc way. Why should this work?

  *The authors have noted that weighted based on distance is more physically intuitive and the authors have redone the analysis with weights based on distance. The weights are defined based on a Gaussian distribution where the closest turbines have the largest weights and the farthest turbines have the smallest weights. This is now defined in Section 3.1. In addition, the authors have also detailed the automatically handling of biases in the measurements in an iterative way. The key being that the data is still used but there is assumed to be a potential bias/error in the measurement that is not being*

*quantified by b. This has previously not been applied in this field and wind direction estimation based on SCADA data continues to be a challenging topic in this field.*

- What is the effect of $w$'s on the results?

  *As mentioned previously, the authors have redone the analysis with weights that depend on distance. The results improve when defining the weights in such a way, which was expected; however, the conclusions remain the same.*

- The most trivial solution is to use a spatial filter. Why not create a simple spatial filter. (There are different extremely simple implementations possible. 1. average of all of the turbines, 2. average of a cluster and use the distance to a central point as a weight.) These trivial solutions need to be explored and compared with the proposed method.

  *The authors acknowledge that there are more straightforward ways to compute the wind direction at each individual turbine. Based on the reviewer's suggestion, the authors have addressed three additional ways that one could compute the wind direction based on additional information from nearby turbine: 1) average across all turbines, 2) weighted average across all turbines based on distance from the turbine, and 3) averaging a cluster of turbines with a weighted wind directions based on distance. The results are shown in Section 5. The results suggest that in most cases other methods are faster; however they are not reliable in the presence of faults/biases in wind direction sensors. Consensus works on the premise that MOST of the sensors are working correctly and it expects that a few sensors are not working properly. Figure 2 shows that when a fault occurs and averaging is invoked across turbines via one of the three methods, the error spreads to other nearby turbines. In the case of averaging across turbines, the error is distributed to all turbines.*

- Related to my previous point, isn't it a bit overkill to use the proposed machinery.

  *As stated previously, the proposed approach is overkill if all of the turbines are operating properly and the corresponding sensors do not have any faults/biases/etc.; however, it is impossible to know without checking all of the turbines 'by hand'. The proposed method provides a layer that systematically checks the turbines against each other to determine potential biases in the data. A bias is quantified in this case and is still used throughout the optimization. The results are shown in a simple wind farm and are then shown on a simulated data set for the larger real-world wind farm.*

- Section 3.2 can be skipped. Just state that you use ADMM to solve the problem.

  *The authors acknowledge that the section may be a bit longer than is necessary and have cut significant portions of it. However, the iterative nature of ADMM is a new approach to estimating wind direction in this field and should be presented to demonstrate the usefulness of an iterative approach and demonstrate the benefits in a feedback-like approach to estimating wind direction.*

- I would also suggest to include a simulation study in which the sensitivity of the proposed method is explored.

  *The authors have now included a sensitivity study on the cluster approach as well as the $\lambda$ parameter, see Figure 5.*

MINOR COMMENTS:

- In the abstract, the authors should add more details regarding the methodology they use.

  *The abstract has been rewritten to include more information on the consensus-based, iterative approach that the authors are using to estimate the wind direction.*

- Pg 2, line 5, I believe that modern wind turbines also employ estimators to get an estimate of the wind speed and wind direction.

*This is now addressed in the introduction."Some turbine manufacturers have wind speed and wind direction estimators to correct for these errors based on individual turbine measurements."*

- Section 2 can be shortened. The information density is rather low.

  *Section 2 has been significantly shortened and only addresses the fact that a wind farm can be represented as a network of agents which is critical for this work.*

- Pg 7, rho is not defined in (4)-(5) but suddenly appears in the experimental section. I believe it is a tuning variable of the ADMM algorithm and I don't understand why it should be tuned for the experiment - Check consistency of the literature list.

  *$\rho$ is defined as a penalty parameter in (6) which enforces the constraints of the problem. This is standard for the augmented Lagrangian with the only stipulation that it has to be greater than zero.*

**2 Reviewer 2:**

The paper proposes an algorithm to estimate the wind direction using a consensus algorithm within relatively small clusters of turbines ( 10 turbines) in a wind farm. The proposed algorithm is based on available SCADA measurements of wind direction at the turbine level. The proposed algorithm does not rely on a physics-based model of the turbines or the flow field.

   The use of consensus algorithm for this application is interesting. The presentation obscures the contribution. There are high-level issues that start with the title and continue with the presentation of certain material. There are technical questions to be addressed. Lastly, there are minor issues that would need to be considered by the authors.

HIGH-LEVEL ISSUES:

- The title does not seemed focused on the problem considered. That is "wind direction estimation using SCADA data" is the topic of the paper. Thus, why "mention autonomous wind farms? In fact, what do that authors mean by an autonomous wind farm?

  *The title has been changed to 'Wind Direction Estimation Using SCADA Data with Consensus-Based Optimization.' Autonomous wind farm was in reference to the automatic decision making in a wind farm at the plant-level and operating as one unit rather than its individual parts. This technology currently is not typically implemented in existing wind farms.*

- One might guess that the answer is in section 2, but this section is not that relevant to the paper; an autonomous wind farm is loosely defined as one that "self-organizes into groups, monitors, and controls its performance in real-time based on existing SCADA data." But this paper has no self-organization and no control.

  *Section 2 has been shortened and a lot of detail has been taking out. In addition, the authors agree that the mention of an autonomous wind farm obscures the main point. Rather the wind farm is taking advantage of the network-like structure of the wind farm and the available data in a wind farm to make real-time decisions. Instead of referring it to as autonomous, the authors have changed the terminology to "collective."*

- Section 2 then goes onto discussing directed and undirected networks. What is the point os discussing directed networks when only undirected ones are used later? This distracts from the main point of the paper. Another point of distraction is that title of Section 2.3; there is no control in this paper.

  *The authors acknowledge this confusion and have removed the section about directed networks and have changed the term "controls" to "operation".*

TECHNICAL ISSUES:

- Elaborate on the selection of rho and lambda. Are these parameters the same for all 10-turbine clusters in the farm?

  *Additional text has been added to this section. Specifically, the authors have added language that specifies that $\rho > 0$ is a penalty parameter that is used ot enforce the constraints of (4)-(5). In addition to $\rho$, there is a penalty parameter, $\lambda$, that dictates the amount of consensus across the wind farm, i.e. a small $\lambda$ encourages larger differences between turbines where as a large $\lambda$ forces nearby turbines to have similar wind directions. This parameter is tuned based on the topology and the amount of wind direction change seen across the wind farm. In summary, $\rho$ is there to ensure that the constraints are met and $\lambda$ is there to enforce the user-defined level of consensus. A sensitivity study to tune $\lambda$ is shown in Section 5.2. A large $\lambda$ is associated with less variability across the wind farm.*

- How do you use the SCADA data to "interpolate" wind direction at the two met tower locations?

  *Due to the proprietary nature of the met tower locations, the exact locations cannot be shown. The location of the sodar is near the outer edge of the wind farm. The wind direction is determined at each of the turbines and the $(x, y)$ locations of the turbines are used to interpolate the wind direction at the location of the sodar.*

- In section 5, you validate the approach at the sodar location (figure 5). This is a good idea and it would helpful to know the sodar location.

  *Due to the proprietary nature of the sodar location, this cannot be shown. However, we can say that the location of the sodar is near the outer edge of the wind farm.*

- After comparison with the sodar, there is comparison between estimated wind direciton and SCADA measured direction. You use consensus amongst 10 turbines to mitigate errors at the single turbine SCADA measurement. Thus, this is confusing to me. Can you clarify? Can you present the uncertainty in the SCADA data? How does the result of the consensus algorithm compare with averaging the wind directions obtained from some of the turbines in a cluster (which is simpler)?

  *Section 5 shows results of different averaging methods to determine the wind direction at an individual turbine. To evaluate these algorithms, the wind farm was processed with simulated data based on the average change in wind direction across the farm and the average standard deviation across the farm. A snapshot of simulated data is shown in Figure 3b and a snapshot of actual data is shown in Figure 7a. This provides a set of truth data to compare the different methods against.*

- It would help to review the literature from the meteorology community on the topic of wind direction estimation from multiple sensors.

  *Additional text is added to the introduction: "Other remote sensing techniques have been proposed as well including radar, lidar, sodar, etc. (Pea et al. (2015); Barthelmie et al. (2016)). However, they all require additional sensing equipment and integration into turbine controllers."*

MINOR ISSUES:

- Equation (8), check the variable being optimized.

  *Updated.*

- What do you mean by $x$ being non-convex after equation (11)?

  *This statement has been removed. Although $x$ remains convex, $b$ is no longer convex and needs an iterative approach to solve for this.*

- Where are the sodar and the met towers in Figure 2a? (4th sentence in paragraph one of section 4.)

  *The met towers and sodar locations are proprietary and cannot be disclosed. The statement was removed.*

**3   Reviewer 3:**

This approach is an interesting alternative to make use of the information from the neighbouring turbines to correct the wind direction (WD) signal from SCADA at the turbine locations. Although it promises valuable contribution to the often overlooked messy WD data processing, it could benefit from a more thorough investigation to study the sensitivity of the developed consensus to some of the local inflow characteristics.

COMMENTS:

- p.1: line 9-10 (Abstract) - "Oftentimes, measurements made at an individual turbine are noisy and unreliable" is rather a blanket statement and it is not clear if only the WD signal is referred to or not. It should be noted that SCADA does/might provide some crucial information, and used in many applications in WF operation, so a re-wording is suggested.

  *The authors agree and have additional language: "... Some turbine manufacturers have wind speed and wind direction estimators to correct for these errors based on individual turbine measurements. Individual measurements, on their own, can be unreliable due to the complex flow created as the wind passes through the rotor, preventing accurate inputs into the individual turbine yaw controller."*

- p.1: line 11 (Abstract) - By taking the nearby turbines into account, the WD signal at an individual turbine can be improved, however, it is equally important to not lose the local information in certain applications. This remark will be repeated several times in these comments.

  *The authors note several times throughout the paper that it is crucial that local information is not lost. This particular approach attempts to smooth the wind direction signal; i.e. wind turbines are not chasing a localized wind gust. In addition, wind turbines should use local information to infer the wind direction under the assumption that the wind direction does not vary significantly between turbines spaced closely together. Variations of these statements are now included in the text.*

- p.2: line 3-4 - What is actually meant by the unnecessary yaw movements? This should be elaborated further... if it is the turbine responding "too fast" to the WD changes, then it can be argued that a simple low-pass filter might be enough. However, in the field it is generally the opposite effect as the turbine generally "too slow" to respond to the highly variant WD (due to its inertia) which is one of the factors that might affect the behaviour of the power curve for highly turbulent flows for example.

  *As stated previously, this refers to the smoothed the wind direction signal could minimize turbines chasing local wind gusts. Language is added to the Results Section that says: "The output of the consensus algorithm shows a smoothly varying wind direction across the wind farm. One implication of smoothly varying wind direction is that it may may reduce the yaw motion of the yaw controller and the yaw drive in that turbines are not chasing local wind gusts that only last for a short time. " In addition, the consensus-based approach is compared with several other averaging techniques, i.e. spatial filters. The results are shown in Section 5*

- p.3: line 1-2 - "...facilitate wake steering WFC..." and more - the benefit of having a reliable WD signal is important for any kind of wake modelling really, including other WFC scenarios, operations management and conditions monitoring.

  *The authors have added this to the introduction and now reads as: "This wind direction estimate can be used as an input to a turbine yaw controller, facilitate wake steering wind farm control (Fleming et al.*

*(2014a)) and other forms of wind farm control, inform operations management, and provide condition
monitoring. It is important to note that this approach requires no additional sensing information."*

- p.4: line 6-7 - It is very true that the turbines that are several kilometers apart would experience highly
different local WDs, however, it could also be the case when the turbines are much nearer. It should
be clarified how near is nearby (possibly depending on local flow conditions) and how much of the local
characteristics are kept and how much of it is smeared among a larger area.

  *The authors agree that an analysis of the number of turbines to include in the consensus-based approach
  is wind farm dependent. An analysis in Section 5, Figure 5, shows this sensitivity to the number of
  turbines to include in the wind direction estimate. In particular, the Figure shows that including too
  few of turbines results in high errors due to the lack of spatial information gained by too few of turbines.
  Similarly, as the reviewer suggests, having too many turbines smears the effects of the wind direction
  across the terrain and also results in a larger error. In this case, communicating with the nearest 15
  turbines produces the best results.*

- p.5: line 13-14 - "...based on nearest 10 turbines" Is there any particular reason for the selection? As
indicated in the previous comment, how much of the local information is intended to be kept in such
a network should be indicated and the reasons should be argued. (e.g. Figure 2b shows some of the
connected turbines are much further apart than the others in the same local network. It is hard to
argue that the contribution from those turbines should be included especially with the same weight,
as will be mentioned later)

  *As mentioned in the previous comment, an analysis was done to determine the clustering size. In
  addition, the weights are now determined based on distance. To the reviewer's point, turbines farther
  away should have less weight than the turbines closer to the turbine of interest. This is now mentioned
  in Section 3. The weights are based on a normal distribution where the closer turbines have a higher
  weight.*

- p.6: line 6-7 - "The objective function, $f_i(x_i)$..." sentence should be omitted as it is confusing compared
to equation (2).

  *This has been removed.*

- p.7: line 2-3 - Why the weight is equal to 1? Especially given that the correlation between some of the
turbines would be lower than others, simply due to distance and local terrain differences.

  *As indicated previously, the weights have been updated to be determined by a normal distribution,
  i.e. closer turbines have a higher weight than farther turbines.*

- p.8: line 24-25 - How is lambda and rho related to/different than the weight $w_{jk}$? Their explicit
definition and tuning procedure should be included.

  *$\rho$ is a penalty term on the constraints. This terms is set to $\rho > 0$ and is not tuned in this proce-
  dure. $\lambda$ is a penalty on consensus and determines the level of consensus across the wind farm. If $\lambda$
  is small, this does not encourage consensus on the wind direction across the wind farm. If $\lambda$ is very
  large, this encourages total consensus across the wind farm. A sensitivity on the $\lambda$ parameter is shown
  in Figure 5a. It shows that high errors occur at small and high $\lambda$ values indicating that some level of
  consensus is required, but not total consensus.*

- Overall Section 3: - It should be discussed as to why this rather complex methodology is/should be
selected over an educated but simple interpolation (maybe combined with a low pass filter if the noise
is a serious concern).

  *Alternative methods are addressed in Section 4 and compared with the consensus-based approach in
  Section 5.*

- p.9: Figure - The met towers should be visible in Figure 2 to understand the validation cases.

  *Due to the proprietary nature of the locations of the met towers and other equipment on site, these locations are not allowed to be disclosed.*

- p.10:line 16-17 - The error sensitivity to wind speed (WS) might be due to the turbulence intensity (TI behavior) with respect to WS. Another color coding or, in general, an additional sensitivity analysis to TI at the sodar would tell a more detailed story.

  *The authors acknowledge that the turbulence intensity is typically higher with lower wind speeds. This is likely the cause. Additional causes could be due to the turbine turning on and off during cut-in wind speeds. The full SCADA data set was not available to the authors; only wind speed, direction, and power. Additional insight into the SCADA data could be helpful to determine the root cause of this. This will be a source of future work.*

- p.11: Figure 4 - Is it also a "snapshot" or an average over some period? Again, one could argue that in 'real-time' we would/might expect higher variance over a relatively big terrain.

  *The data provided in Figure 7a (used to be Figure 4) is a 1 minute average of data. The wind speed is approximately 6.5 m/s in most locations. The authors agree that higher variance is present at lower wind speeds. The authors are currently working on incorporating wind speed into this consensus-based approach, but will not be included in this paper.*

- p.11: line 2 - Again, due to the turbine's inertia, we generally see the opposite effect in fact - turbine is having a hard time to catch up with (truly, not just noisy) variable WD sometimes. Therefore, it is generally expected the (non-intentional) misalignment to occur due to the bias in the sensors, rather than the noise.

  *The authors agree that there is a large lag time in yaw controllers to ensure the wind direction has sufficiently changed. However, this study compares the wind direction signal with the estimated wind direction signal rather than a yaw error. The authors note the confusion in the wording of the paper and have updated the language to indicate wind direction error rather than yaw error. This is noted in the results section.*

- p.11: line 3-4 - An important factor in terms of defining the true error might be the difference between the wind vane measurements and the actual nacelle position (as mentioned very briefly later in the paper during the filtering of the data for the last validation case). In most of the turbines' SCADA, there exist a separate signal (than the wind vane measurements, generally called wind direction) which is called nacelle position or nacelle direction. Those two signals differ quite a bit, especially in high turbulence cases (turbine not following the highly variant WD as mentioned earlier here). It is important to clarify what is 'corrected' in this study is only the wind vane measurements which may differ from where the turbine is actually facing.

  *The authors note the confusion and have compared the SCADA data wind direction signal (yaw position + wind direction) to the consensus-based wind direction. This is noted in several locations in the paper.*

- p.13: Figure 6 - The effect of the changing TI levels between the two yaw error cases defined in Figure 6a should be clarified.

  *The turbulence intensity levels were not investigated in this study. However, this will be a focus of future work.*

- p.13: line 13 - Suggest rewording to "truth": reference, baseline, true value.

  *The authors have reworded "truth" to reference.*

- p.13: line 13-14 - not clear how and why the error estimates from the consensus are more reliable than the sodar measurements? Depending on the equipment itself surely, sodar has been shown to agree very well with the met mast measurements on the site (e.g. Steven Lang, Eamon McKeogh (2011) LIDAR and SODAR measurements of wind speed and direction in Upland Terrain for Wind Energy Purposes, Remote Sensing)

    *The authors agree that the sodar is able to provide a reliable wind direction measurement. This is why the authors use the sodar as a reference measurement when comparing the consensus-based wind direction measurement. The main point of (now) Figure 11 is to demonstrate that one single point of wind direction measurement is not able to capture the wind direction errors across the wind farm. To assess the wind direction errors at turbines across the wind farm, the consensus-based wind direction signal is able to determine the small and large errors experienced at each turbine.*

- p.14: Figure 7 - The difference between the consensus and the wind vanes seem to increase with increasing "immediate wake" effects on top of the nacelle, again pointing towards the (added) turbulence sensitivity. Overall, the effect of turbine not being able to follow the highly fluctuating WD, hence dynamic misalignment and under performance, is more of a physical phenomenon due to turbine inertia. Correcting the signal would not necessarily solve that problem.

    *The authors acknowledge that this is definitely possible. The authors are working to link a wake model to this study to further understand the performance of wind direction estimation. However, additional data and simulations will need to be conducted to determine the errors that can be addressed with consensus-based control. The authors are optimistic given these results, but further work is required.*

- p.15: line 7-9 - 'Smearing' the small timescales in local WD might be useful for many other analysis; however, if the turbine avoids yawing as a result, we might risk to lose power still. Correcting the bias in the signals on the other hand, is a much more useful outcome of such approach.

    *The authors agree that correcting a bias is a useful result. Biases are automatically detected with this algorithm and are detailed in Section 3.*

---

## Author Response (AR2)

**Review Comments for "A Framework for Autonomous Wind Farms: Wind Direction Consensus"**

Corresponding author: Jennifer Annoni

May 13, 2019

**Abstract**

The authors would like to thank the reviewer for their comments. The minor changes have been made based on the reviewer's suggesions.

**1 Reviewer 2:**

- Thanks for the clarification and additions made to the article. The revised version of the manuscript is much improved. Although the very relevant discussion on turbulence sensitivity remains lacking, I hope that the future improvement of the approach will include that investigation. For this manuscript, I suggest to add it to the Future Work/Conclusions Section.

  *The authors have added a note in the Future Work/Conclusions Section.*

- Page 9, (regarding Figure 2): The definition of the error term here should be stated (for mean in this Figure, and Max in Table 1 later in the text), especially since seemingly 90 deg error gives 15 deg error for the top left plot of the sensors. If it is a "representative error" for the whole wind farm (i.e. individual error divided by number of turbines) the reasons for that representation should be discussed. For the proposed use cases of the presented method (wake steering, faulty WD signal detection, etc.), it is important to know which turbines within the wind farm have erroneous signals. This is because some erroneous WD signals are more crucial than the others within the wind farm for e.g. wake modeling.

  The definition of this error also leads to a peculiar behavior comparing the averaging methods. For example, although the weighted average has 3 out of 6 turbines matching the "True" value, it still has a higher error than the Equal average which has all the turbines pointing to a different direction.

  *Additional text has been added. The maximum error has similar trends to the mean absolute error. The mean absolute error becomes a more "representative metric" when looking across a larger wind farm that has noise as shown in Section 5.2. Added text:*

  *"The mean absolute error shows that these other methods have the ability to reduce the "average" error across the wind farm. This metric is important when assessing the accuracy of a method across a wind farm as will be shown in a larger wind plant example in Section 5.2. However, in this example, there is only one fault/error and the plots in Figure 2 show that the error has spread to more turbines. In this case, it is critical that the consensus algorithm is able to identify the erroneous wind direction signal and minimally impact the other turbines in the wind farm. This will have implications when implementing advanced wind farm control strategies like wake steering. "*

- Page 12, lines 5-8: It is seen in Figure 6 error comparison that, the performance of the simpler methods (particularly Cluster averaging, as can be expected) are similar to the proposed method. This similarity and the advantage of the proposed method (under biased and/or faulty signals) is very briefly discussed here. Can we then say that it might be recommended to use cluster averaging approach for non-biased or corrected data if simplicity and computational efficiency is a concern?

  *The authors think that is a fair conclusion to make and is now stated in the text.*

- Page 15, lines 18-20: Since there is one offset presented for the whole wind farm at Figure 10(b), should we read it as the individual offsets (which are varied across the wind farm) are normalized by the number of turbines? If that is the case, then it is more likely that a limited number of turbines would be misaligned for longer periods, instead of the whole wind farm being misaligned for more than 50% of the time. The sentences should be rephrased to clarify this.

  *Additional text is added to reflect this point that it could be only a handful of turbines misaligned for long periods of time: "This plot indicates that some turbines across the wind farm could be spending a significant amount of time misaligned."*

- Page 16, - Figure 9: Would be nice to see the error distribution over the investigated period - possibly with the filtered wind speed (e.g. for wind speed ¿ 5 m/s) to see the temporal statistics of the error.

  *An error distribution has been added to the text as Figure 10 for wind speeds greater than 4.0 m/s.*